# Research on Clamping Action Control Technology for Floating Fixtures

**DOI:** 10.3390/ma15165571

**Published:** 2022-08-13

**Authors:** Benchi Zhu, Zhuang Mu, Wenbo He, Longxin Fan, Guolong Zhao, Yinfei Yang

**Affiliations:** 1College of Mechanical and Electrical Engineering, Nanjing University of Aeronautics & Astronautics, Nanjing 210016, China; 2School of Mechanical Engineering, Changzhou University, Changzhou 213164, China

**Keywords:** floating clamping, machining deformation, clamping action control, strain energy evolution

## Abstract

By adaptively releasing deformation during machining, floating clamping significantly raises the machining quality of aircraft structural parts. The fundamental issue to be resolved is how to precisely control the clamping action of the floating fixtures. In this study, the machining process of aircraft beams was studied, utilizing the finite element method (FEM) from the perspective of strain energy evolution. The study found that the increment of deformation and the variation of the strain energy between adjacent removed layers of the material showed the same trend of change, and targeted clamping loosening at the stage of an excessive strain energy evolution gradient is beneficial to reducing the final deformation of the workpiece. Therefore, a clamping action control method based on strain energy evolution gradient regulation is proposed, and a clamping action control strategy of floating fixtures was formulated. Furthermore, a cutting experiment was carried out, and the results showed that the maximum deformation of the aircraft beam using the clamping action control strategy was only 0.112 mm, which was reduced by 74.6% compared to traditional clamping.

## 1. Introduction

Aluminum alloy large-size structural parts are widely used in aerospace manufacturing due to their light weight, high strength, and long fatigue life [1]. The application of large-size structural parts in the wings and fuselage of an aircraft can increase its strength and maneuverability while reducing its weight. Large-size structural parts are usually formed by milling. However, these parts are large in size (the length is more than 1 m), thin in wall (the minimum thickness can reach 1 mm), and complex in structure, which is typical for difficult-to-machine parts, and their material removal rate is often above 90%. As the material is removed, the workpiece is easily deformed, because its rigidity gradually becomes weaker [2,3]. Therefore, the machining specifications of the workpiece are difficult to meet. For example, the thin-walled beam of a certain type of aircraft has a size of 6000 mm × 400 mm × 100 mm, and the machining deformation is often 2 mm, but the accuracy is only required to be ±0.3 mm [4]. It can be seen that the difficulty of machining deformation control of large-size structural parts is increasing, which is an urgent problem that needs to be solved at present [5,6].

The continuous improvement of clamping technology is conducive to solving the problem of the machining deformation of large-size structural parts [7,8]. De Meter et al. [9] proposed the fast support layout optimization (FSLO) model for increasing workpiece rigidity to reduce deformation. Shi et al. [10] designed a local surface damper to reduce the machining deformation and vibration of thin-walled workpieces by enhancing the rigidity of the clamping system. Munro et al. [11] proposed a flexible clamping method based on an array column for large-size skins, which can adapt the shape of the workpiece by controlling the expansion and contraction of the column and uses a vacuum chuck to achieve stable clamping of the workpiece. Vishnupriyan et al. [12,13] analyzed the location error source of the workpiece and optimized the clamping parameters, utilizing the FEM and a genetic algorithm (GA), thereby reducing the deformation on the entire machining path. However, the above-mentioned methods and devices cannot realize the clamping adjustment during machining, so it is difficult to meet the requirements of deformation control of large-size structural parts.

At present, the combined application of high-performance adaptive clamping units and advanced computer numerical control (CNC) machining technology has a significant effect on the deformation control of large-size structural parts [4,14]. Nee et al. [15] proposed an intelligent fixture for thin-walled part machining, which can provide a precise and variable clamping force. Möhring et al. [16] developed flexible clamping unit integrating sensors and floating fixtures, which can control the rigidity of the system by adjusting the clamping force and compensate for machining deformation by adjusting the pose of the fixture. Yu et al. [17] proposed a double spherical fixture with eight degrees of freedom for the machining of thin-walled blades, which can reduce the deformation of the workpiece through stress-free clamping and stress release during machining. Bakker et al. [18] used a proportional integral (PI) controller to realize the adaptive adjustment of the clamping force to adapt to the change in the force between the workpiece and the fixture during machining, thereby controlling the deformation of flexible workpieces. Li et al. [19] proposed an adaptive machining method based on the responsive fixture, which adjusts the position of the fixtures in real time according to the measured deformation of an online measurement system.

As a new type of clamping technology, floating clamping provides a new idea for the machining deformation control of workpieces. The floating clamping method lifts the workpiece away from the worktable by multiple clamping units and regulates the clamping action according to the machining conditions of the workpiece during machining, so that the workpiece is in a state of lower stress potential energy, thereby achieving the purpose of reducing the deformation. Compared to the traditional clamping method, the biggest highlight of floating clamping is that the threshold for triggering clamping loosening can be reasonably set during machining. On the premise of ensuring the locating datum of the workpiece, some fixtures are allowed to perform the loosening action to release the deformation, and iteratively eliminate the deformation released in this part of the process through the next machining [16,19].

However, in practical applications, how to accurately control the clamping action of floating fixtures is the main technical problem of floating clamping. During machining, too many deformation releases affects the machining efficiency of the workpiece; if the number of deformation releases is too small, the residual stress accumulated in the workpiece cannot be fully released, so the deformation of the workpiece cannot be effectively reduced. Therefore, formulating a reasonable clamping action control strategy is the key research issue in floating clamping. At present, most of the clamping action control technologies for traditional fixtures rely on the sensor monitoring of force and displacement, while studies on clamping action control technology for floating fixtures, at home and abroad, are few. Therefore, the existing floating clamping technology is not mature.

From the perspective of energy, this research studied clamping action control technology for floating clamping, utilizing calculation and simulation. By analyzing the evolution process of strain energy during workpiece machining, we could identify the stage where the gradient of strain energy evolution was too high, and then trigger the clamping to loosen in a targeted manner. Therefore, we propose a clamping action control strategy for fixtures suitable for floating clamping machining and verified the effectiveness of the strategies through experiments.

## 2. Deformation Release Principle of Floating Clamping

From the perspective of energy, different material removal strategies for workpieces produce different boundary conditions, which have a great impact on the elimination of internal energy and work energy conversion of the workpiece during machining.

With the continuous removal of blank materials, the energy contained in the blank drops until it is finally formed. However, the strain energy contained in each material removal layer of the same thickness inside the workpiece is different; that is, the strain energy density of each removed layer is not completely consistent, which makes the decreased gradient of the strain energy inside the blank during the whole machining process different [20]. According to the principle of work energy conversion, with the progress of machining, the elastic strain energy inside the workpiece is transformed into work after loosening the clamping and the work is released in the form of deformation; while the workpiece cannot be deformed when the fixture is kept clamped, this part of the strain energy is not converted into work, but exists inside the workpiece to cause deformation, which is the essential reason for the deformation of the workpiece from the perspective of energy [21,22]. Therefore, the evolution gradient of the internal strain energy of the blank within the machining process affects the deformation of the workpiece to a certain extent. In the stage where the gradient of the strain energy evolution is too high, it produces a larger workpiece deformation or a larger deformation trend than in other stages due to the clamping not being loosened.

Compared to traditional fixed clamping, floating clamping can flexibly trigger the clamping to perform different actions during machining. Relying on this advantage, this paper proposes a clamping action control method for floating fixtures based on the regulation of the strain energy evolution gradient, as shown in Figure 1.

Assuming that the initial elastic strain energy of the blank before machining is $E_{0}$, when the clamping is released after machining, the energy contained in the formed workpiece is $E_{r}$; then, the total energy loss of the workpiece is ΔE, and the strain energy change between step *i* and step *i* − 1 is recorded as $\Delta E_{i}$. By analyzing the evolution gradient of the strain energy within the machining process in advance, the stage where the strain energy of the workpiece drops sharply is identified. At this stage, the step with a larger ΔE is identified, so that the clamping action is triggered before this step in a targeted manner to release the deformation. Taking step *i* as an example, assuming that step *i* is in the stage where the strain energy of the workpiece drops sharply during machining, the elastic strain energy density of the material removal layer corresponding to step *i* is higher, and $\Delta E_{i}$ is larger when milling to this layer. The traditional clamping method ignores the energy release of the workpiece during the clamping process. After the machining is completed, the energy inside the workpiece begins to release with the unloading of the clamping, which causes a large deformation. However, floating clamping pays attention to the clamping adjustment during machining. The clamping action is triggered before step *i* to loosen the clamping so that the elastic strain energy accumulated inside the workpiece can be released during the machining. At this time, the workpiece releases a certain deformation that can be corrected by the next milling. If the deformation is large, the CNC machining program should be modified according to the theoretical model, and the deformation should be removed by adding a one-step cutting program. It follows, then, that when using floating clamping for machining, the $\Delta E_{i}{}^{\prime }$ generated in step *i* includes the two steps of “loosen clamping” and “follow-up machining”, the strain energy inside the blank drops gently, and the tendency for spring-back deformation inside the workpiece is eliminated in the machining gap. Finally, a formed workpiece that meets the requirements of machining accuracy is obtained.

According to the above deformation release principle, a clamping action control strategy for floating fixtures suitable for floating machining can be formulated by deeply studying the workpiece machining deformation with the help of the FEM. Based on this, the PC determines whether to issue a command to trigger the clamping to loosen. The control flow of the floating clamping machining is shown in Figure 2.

## 3. The Energy Deformation Evolution of the Workpiece

### 3.1. Analysis of Beam Workpiece Deformation Based on Energy Principle

This section takes the aircraft beam as an example to analyze the evolution of the strain energy of the workpiece during machining. Generally speaking, the overall deformation of beam parts after machining is regarded as elastic deformation. Under the action of external force, the elastic structure generates strain energy inside, and its internal strain energy can continue to do work after unloading all the force. Assuming that the stresses in the three directions of X, Y, and Z on the micro-element inside the structure are $\sigma_{x}$, $\sigma_{y}$, and *σ_z_*, respectively, the internal strain energy *U*_1_ and the strain energy density μ in the blank are:(1)$$U_{1}=∭\frac{1}{2E}\left(\sigma_{x}^{2}+\sigma_{y}^{2}+\sigma_{z}^{2}\right)-\frac{V}{E}\left(\sigma_{x}\sigma_{y}+\sigma_{y}\sigma_{z}+\sigma_{x}\sigma_{z}\right)dV$$
(2)$$\mu =\frac{1}{h}\int_{0}^{t}\frac{\sigma_{x}^{2}+2\nu \sigma_{x}\sigma_{y}+\sigma_{y}^{2}}{2E}dz$$
where E is the elastic modulus of the material, ν is the Poisson’s ratio of the material, and h is the thickness of the blank.

After layer-by-layer milling, the remaining part of the energy in the workpiece realizes the conversion of work and energy, and the residual energy is redistributed inside the workpiece to achieve a new balance of force and torque, thereby causing the deformation of the workpiece. Assuming that the initial residual stress inside the blank is symmetrically distributed from the neutral layer along its thickness direction, after the material of layer *i* is removed, the bending bending moment contained in the remaining material of the blank is:(3)$$M_{i}=\int_{Ai}\sigma \left(z-\bar{z_{i}}\right)dA_{i}$$

According to the actual working conditions, the overall deformation of the aircraft beam is regarded as the deformation of the simple supported beam; then, after the material of layer *i* is removed, the elastic deformation of the workpiece is [22]:(4)$$\omega_{i}=\frac{M_{i}L^{2}}{8EI_{i}}$$

As the machining progresses, the energy inside the workpiece drops continuously. After the clamping is loosened, the area where the strain energy is too high or intensive in the workpiece is deformed. According to the first law of thermodynamics, the work energy conversion during workpiece machining can be expressed as:(5)$$\delta E_{k}+\delta U_{1}=\delta W+\delta Q$$
where $\delta E_{k}$ and $\delta U_{1}$ respectively represent the variation of the internal kinetic energy of the workpiece and the variation of the initial strain energy in the workpiece during the energy release; δW and δQ respectively represent the work performed by the deformation release of the workpiece and the heat change of the workpiece. According to the “Conversion Hypothesis of Work and Energy Theory” [23], the transfer of internal heat and the change of kinetic energy are usually ignored in the short process of energy release, so $\delta U_{1}=\delta W$, which proves that the release of the internal strain energy of the blank is the main reason for the deformation of the workpiece from the energy point of view.

### 3.2. Analysis of Strain Energy Evolution Process in the Machining of Beam Workpieces

Based on the above theory, the removal process of workpiece material was analyzed from the perspective of energy with the help of the FEM. The object studied in this research was the thin-walled beam made of 7050-T7401 aluminum alloy, which is machined by layer-by-layer milling. The material compositions are shown in Table 1, and the material properties are shown in Table 2. As shown in Figure 3, the middle part was the final workpiece, the material to be removed was divided into *n* layers, and the thickness of each layer was $u_{i}$. Among them, the value of $u_{i}$ here was 0.05 mm, and the value of *n* was 20.

The blanks of thin-walled beams are generally pre-stretched. By measuring the stress of the blank, the initial residual stress field shown in Figure 4 was obtained by fitting and then input into the model.

The relationship between the strain energy and removal amount obtained by the FEM is shown in Figure 5a. The study found that with the continuous progress of machining, the residual strain energy in the workpiece showed a changing trend of “severe decrease—gentle decrease—severe decrease” to a certain extent. For the aircraft beam studied in this research, when the material removal amount was in the range of 0~0.3 mm and 0.5~0.8 mm, the strain energy in the material was in a severe drop stage. In this process, the stage where the removal amount was less than 0.3 mm is defined as the first severe energy drop stage, and the stage where the removal amount was between 0.5 mm and 0.8 mm is defined as the second severe energy drop stage. However, when the material removal amount of the workpiece was between 0.3 mm and 0.5 mm, the decreasing trend of the strain energy in the blank was relatively gentle, and this stage can be defined as the steady energy drop stage.

Figure 5b shows the increment of workpiece deformation and the variation of the material residual strain energy in each machining step. It can be seen that the changing trend of the deformation between adjacent removed layers and the variation of the residual strain energy are the same. During machining, if a step or several steps make the evolution gradient of the residual strain energy of the material too high, and the residual strain energy drops sharply, the deformation increases sharply. Therefore, it is extremely necessary to reasonably design or improve the clamping strategy for the workpiece to release the internal strain energy or reduce the strain energy evolution gradient. This would make the deformation of the workpiece in each step basically the same and would not cause the workpiece to have a surge in deformation that could not be corrected by subsequent machining after the milling.

## 4. Clamping Action Control Strategy for Floating Fixtures

Through the energy method, Fan et al. [24] calculated the web deformation $d_{i}$ of a beam workpiece when milling to layer *i*. Based on this method, the PC can calculate the deformation of the web in real time during machining. Then, according to whether the deformation $d_{i}$ meets the maximum allowable deformation $d_{a}$ on one side of the web and the single-layer cutting allowance $a_{\max}$, it is judged whether to trigger the clamping loosening action. Among them, $d_{i}$ can be calculated from Equation (4), and the calculation formula of $d_{a}$ is as follows:(6)$$d_{a}=r_{l}+t_{1}-a_{\min}$$
where $a_{\min}$ is the minimum single-layer cutting allowance, *r_l_* is the overall cutting allowance, and $t_{1}$ is the lower deviation of the dimensional tolerance of the web.

From the perspective of energy, in the severe energy drop stage, the gradient of the strain energy drop is too large, and there is a large amount of residual strain energy in the material that needs to be released in a concentrated manner. At this point, it is necessary to perform several clamping loosenings. The average value of the change between the residual strain energy after the machining of each step in each sharp energy drop stage should be recorded as $\bar{\Delta E_{1}}$, $\bar{\Delta E_{2}}$, …. According to whether $\Delta E_{i}$ is higher than the average value of the strain energy change in this stage, it is decided whether to trigger the clamping to loosen.

Combining the above analysis, based on the real-time calculation of the deformation of the aircraft beam during the machining, a control strategy for the trigger clamping action for floating clamping is proposed as follows:(1)Analyzing the evolution of strain energy with material removal utilizing the FEM:
a.When the machining process is in the first severe energy drop stage, if the condition is met that $\Delta E_{i}>\bar{\Delta E_{1}}$:At this time, the clamping should be loosened before the current step, so that the strain energy can be released, so as to reduce the gradient of the strain energy evolution to avoid a sudden increase in deformation that cannot be corrected.b.When the machining process is in the second severe energy drop stage, if the condition is met that $\Delta E_{i}>\bar{\Delta E_{2}}$:The second severe energy drop stage is closer to finishing, and the clamping should be loosened before the current step. At this time, it is necessary to avoid the deformation of the workpiece due to the high gradient of strain energy evolution.c.When the machining process is in the steady energy drop stage, the evolution gradient of the strain energy of the workpiece is small. At this time, all fixtures remain clamped, and the machining is continued to ensure the production of the workpiece.(2)Analyzing the deformation of the workpiece calculated by the PC using the energy method during machining:
a.If the deformation meets the conditions: $d_{i}\leq d_{a}$ & $d_{i}\geq a_{\max}$:At this time, although the workpiece tends to deform, its current deformation is still within the allowable deformation range, and the deformation can be removed by the next cutting process, so there is no need to loosen the clamping; machining can be continued.b.If the deformation meets the conditions: $d_{i}\leq d_{a}$ & $d_{i}\geq a_{\max}$:At this time, although the deformation of the workpiece is still within the allowable range, it cannot be eliminated by the next cutting. Therefore, it is necessary to loosen the clamping at this time to release the deformation of the workpiece, thereby reducing its internal residual energy. The rebound deformation of the workpiece can be completed in a short time. The deformation is released after waiting for 2–3 s, the PC re-issues the clamping command, and each floating clamping module re-clamps the workpiece according to the position and pose of the adjusted workpiece.c.If the deformation meets the conditions: $d_{i}\geq d_{a}$:At this time, the deformation of the workpiece is too large. It is necessary to completely loosen the clamping to readjust the machining datum; otherwise, the workpiece will not meet the machining accuracy requirements.(3)When the last cutting layer is about to be roughed, it is necessary to loosen the clamping to fully release the deformation of the workpiece, and then re-clamp them to make the last correction before finishing machining.

The above process is described in Figure 6.

The clamping action control strategy described above was adopted to re-analyze the machining process of the aircraft beam, utilizing the FEM. Figure 7a reflects the comparison relationship between the variation of the strain energy before and after the application of the clamping control strategy. It can be seen that, compared to not applying the floating clamping control strategy, the amount of change in strain energy during the machining greatly dropped after applying the floating clamping control strategy described in this paper, and the maximum reduction was 76.0%. Figure 7b reflects the variation of the strain energy during the machining of the workpiece before and after the application of the clamping control strategy. It can be seen that the floating clamping control strategy can effectively reduce the strain energy of the workpiece and its evolution gradient of the strain energy during machining, so that the strain energy of the workpiece material decreases steadily and slowly with machining, so as to correct the deformation during the machining at any time and reduce the final deformation of the workpiece.

## 5. Experimental Verification and Discussion

### 5.1. Milling Experiment of Floating Clamping

A typical thin-walled beam was taken as the experimental object, and the clamping action control strategy was applied to the milling. The size of the workpiece was 475 mm × 140 mm × 20 mm, the thickness of the side wall was 5 mm, and the thickness of the web was 3 mm. The specific structure of the workpiece is shown in Figure 8, and the floating clamping system is shown in Figure 9. The blank was made of 7050-T7451 aluminum alloy with a size of 500 mm × 200 mm × 25 mm. The material of the blank was evenly divided into 25 units along the thickness direction, and the thickness of each unit was 1 mm.

According to the clamping action control strategy formulated in Section 4, the specific process route of this workpiece was formulated, as shown in Figure 10:

In this study, two workpiece blanks were machined successively. Among them, the No. 1 workpiece was machined with the clamping action control strategy described in this paper, and the No. 2 workpiece was machined with one-time clamping. This machining experiment was carried out on the NingQing VC-3016G gantry CNC machining center, and the tool used in the experiment was a cemented carbide ϕ16 end mill. A hydraulic pump provided clamping force for the floating fixtures. The measurement of the deformation was performed by a RENISHAW RMP60 probe. The probe was installed on the spindle of the machine tool and kept in contact with the workpiece; the coordinate value of the monitoring point of the workpiece was obtained according to the coordinate value of the machine tool spindle and the specific positional relationship between the contact and the workpiece. The deformation of the workpiece was obtained by measuring the coordinates of the monitoring points before and after the machining was completed. The cutting parameters are shown in Table 3, and the experiment is shown in Figure 11.

### 5.2. Discussion of Experimental Results

Figure 12 shows the positions of the eight deformation monitoring points, and Figure 13 shows the deformation monitoring values of the eight measuring points before and after applying the clamping action control strategy. Among them, the No. 2 workpiece (without using the clamping action control strategy) had a maximum spring-back deformation of 0.458 mm and a minimum spring-back deformation of 0.318 mm; however, the No. 1 workpiece (using the clamping action control strategy) had a maximum spring-back deformation of 0.116 mm and a minimum spring-back deformation of 0.022 mm. Comparing workpiece one and workpiece two, the maximum and minimum deformation values of the workpiece were reduced by 0.342 mm (74.6%) and 0.296 mm (93%), respectively, after applying the clamping control strategy, and the spring-back deformation of each part of the workpiece was significantly reduced.

The results show that the clamping action control strategy of the floating fixtures based on the regulation of the strain energy evolution gradient can effectively improve the machining quality of a thin-walled workpiece and meet the machining quality requirements of the workpiece.

## 6. Conclusions

In this paper, by studying the deformation release principle of the floating clamping and the energy deformation evolution process of the workpiece, a floating clamping action control technology based on the regulation of the strain energy evolution gradient is proposed, and a clamping action control strategy of the floating fixture was formulated. Taking the aircraft beam as an example, the effectiveness of the strategy in reducing the machining deformation of the workpiece was verified through simulation and an experiment. The main conclusions are as follows:(1)The increment of deformation and the variation of the strain energy between adjacent removed layers of the material showed the same trend of change. When the variation of the strain energy increased, the deformation of the workpiece increased; that is, when the evolution gradient of the strain energy was high, the machining deformation of the workpiece also increased sharply. Therefore, a floating clamping action control method based on the regulation of the strain energy evolution gradient is proposed, and a clamping action control strategy of floating fixtures was formulated.(2)After applying the clamping action control strategy, the variation of the strain energy during the machining greatly dropped (the maximum reduction was 76.0%), the energy decreased steadily and slowly, and the final deformation of the workpiece was significantly reduced.(3)The milling experiment of the aircraft beam was carried out by applying the clamping action control strategy. After the machining, the deformation of the workpiece fully met the machining accuracy requirements of the workpiece, and the average deformation was reduced by 74.6% compared to traditional clamping. The experiment effectively verified that the clamping action control strategy is suitable for the floating clamping method, which can effectively reduce the machining deformation of thin-walled beams with weak stiffness, under the premise of ensuring productivity and meeting the needs of industrial applications.

## Figures and Tables

**Figure 1 materials-15-05571-f001:**
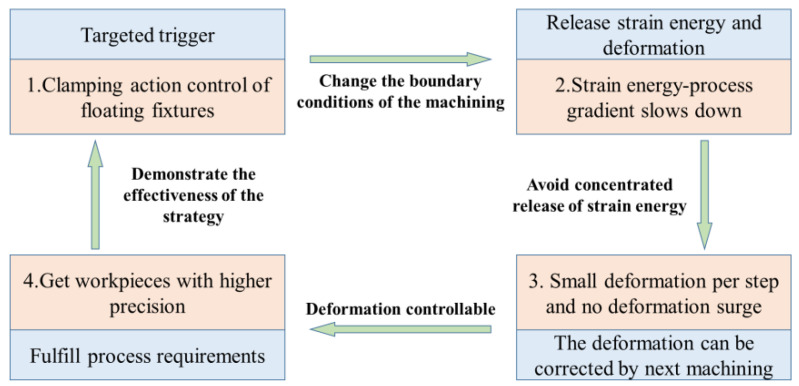
Principle of clamping action control based on gradient control of strain energy.

**Figure 2 materials-15-05571-f002:**
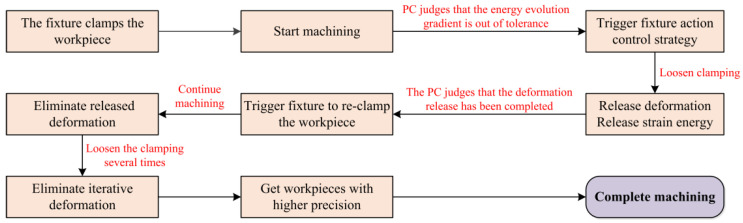
Control flow of floating clamping machining.

**Figure 3 materials-15-05571-f003:**
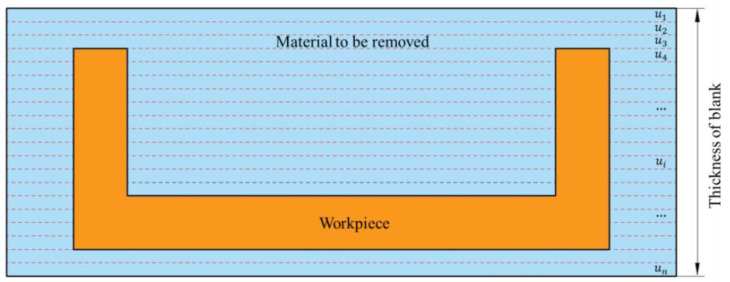
Schematic of material layer-by-layer milling.

**Figure 4 materials-15-05571-f004:**
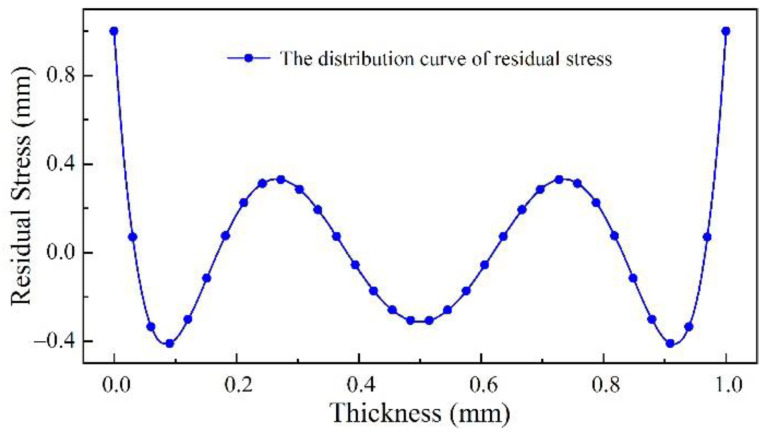
Initial residual stress model.

**Figure 5 materials-15-05571-f005:**
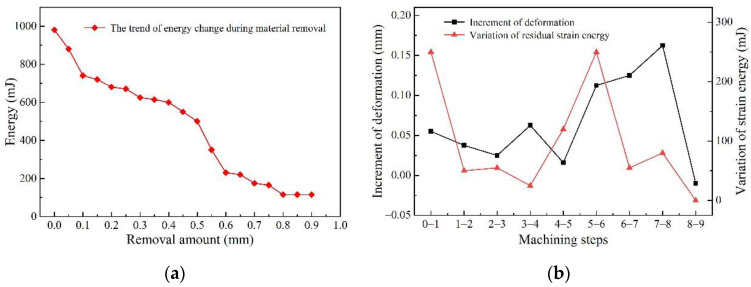
Variation of deformation and strain energy during machining: (**a**) relationship between strain energy and removal amount; (**b**) relationship between the increment of deformation and the variation of residual strain energy.

**Figure 6 materials-15-05571-f006:**
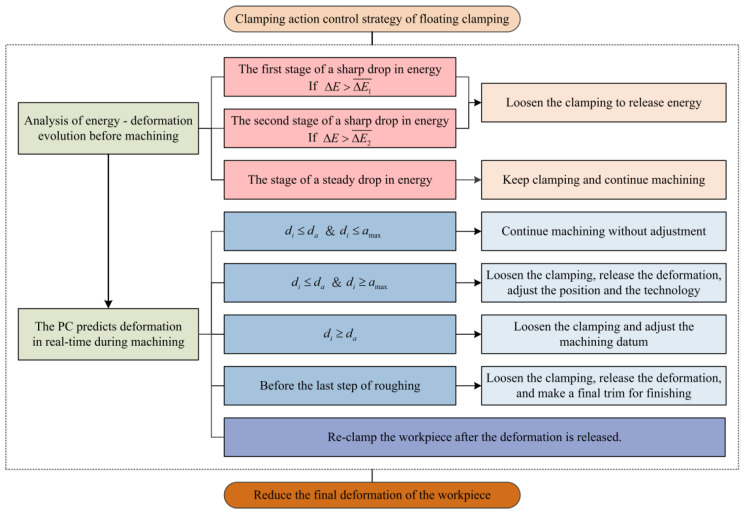
Clamping action control strategy for floating clamping machining.

**Figure 7 materials-15-05571-f007:**
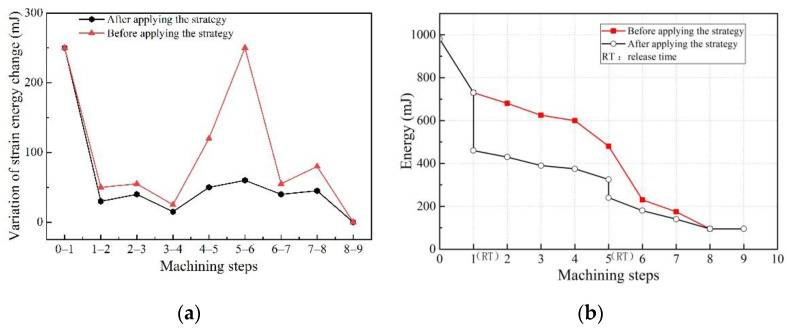
Effect of clamping control strategy on strain energy: (**a**) comparison of the variation of strain energy before and after applying clamping control strategy; (**b**) comparison of strain energy evolution process before and after applying clamping control strategy.

**Figure 8 materials-15-05571-f008:**
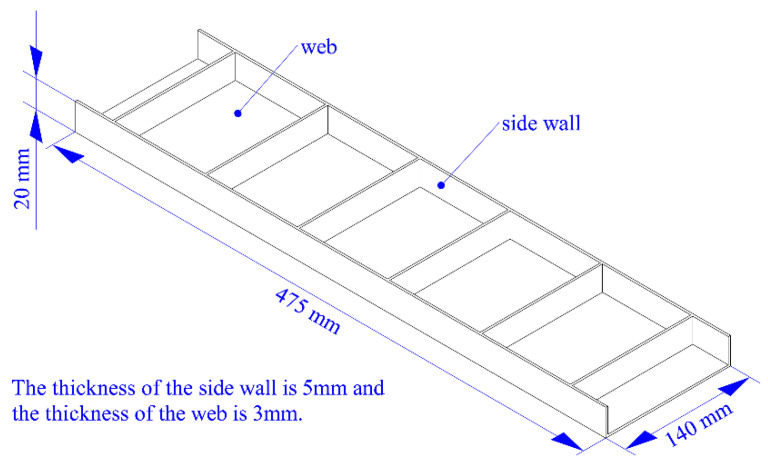
The structure and size of the thin-walled beam.

**Figure 9 materials-15-05571-f009:**
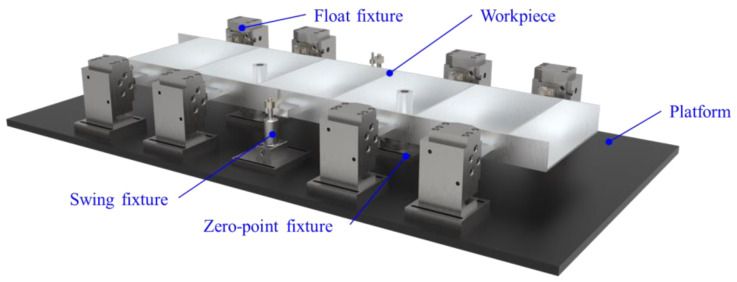
Floating clamping system.

**Figure 10 materials-15-05571-f010:**
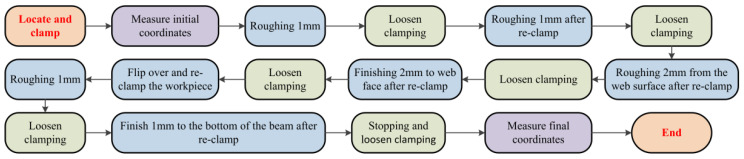
The process route of workpiece machining.

**Figure 11 materials-15-05571-f011:**
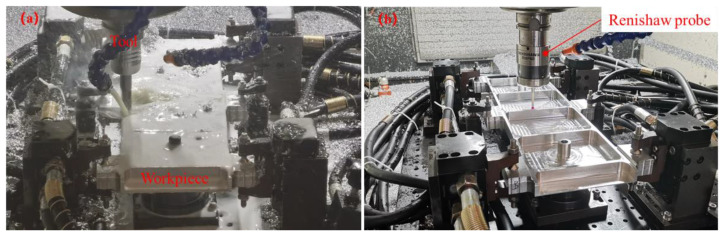
Experiment and measurement: (**a**) milling operation; (**b**) measuring deformation.

**Figure 12 materials-15-05571-f012:**
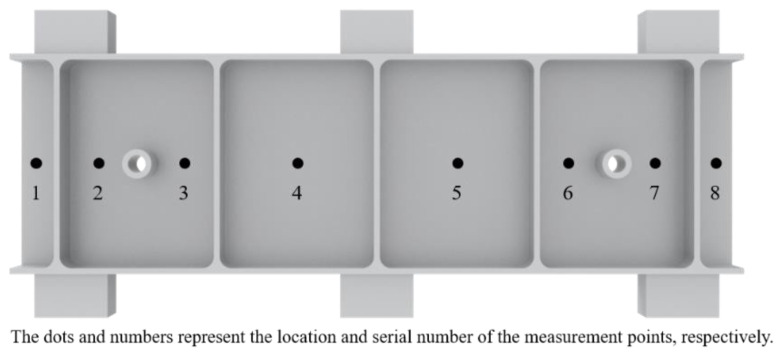
Distribution of deformation monitoring points.

**Figure 13 materials-15-05571-f013:**
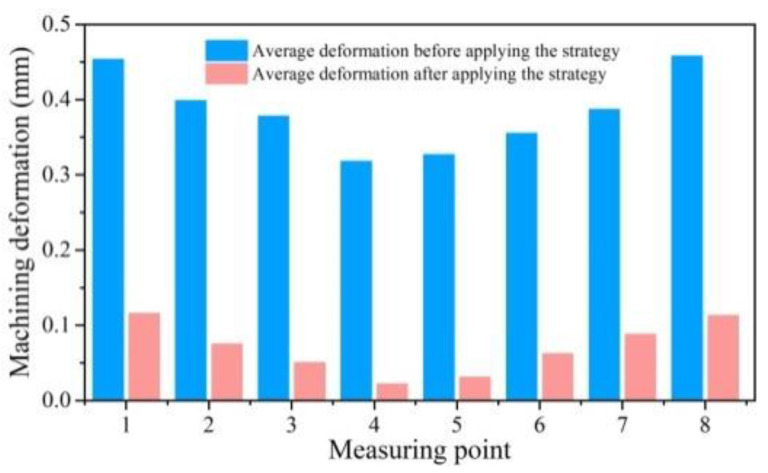
Comparison of the deformation before and after applying clamping control strategy.

**Table 1 materials-15-05571-t001:** Chemical composition of 7075-T7401 (%).

Si	Fe	Cu	Mn	Mg	Cr	Zn	Ti	Etc	Al
0.4	0.5	1.2~2.0	0.3	2.1~2.9	0.18~0.28	5.1~6.1	0.2	0.15	margin

**Table 2 materials-15-05571-t002:** Mechanical properties of 7075-T7401.

Elastic Modulus (GPa)	Yield Strength (MPa)	Poisson’s Ratio	Density (kg/m^−3^)	Hardness (HB)
71	455	0.33	2.81	150

**Table 3 materials-15-05571-t003:** Cutting parameters [2].

Cutting Parameters	Parameter Value
Cutting depth *a_p_* (mm)	1
Cutting width *a_e_* (mm)	6
Spindle speed *n* (rpm)	8000
Feed speed *f* (mm/min)	4000
Tool diameter (mm)	16
Tool tooth number *N*	3

## Data Availability

Not applicable.

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
