# Peer review of "Research on Clamping Action Control Technology for Floating Fixtures"

_materials, 2022, doi:10.3390/ma15165571_

Round 1

Reviewer 1 Report

Review of the manuscript entitled "Research on clamping action control technology of floating fixtures".

This is a poorly written review paper, which is difficult to follow and hard to understand. I do not think this manuscript is qualified to published on this journal, consequently I can’t recommend this paper for publication.

In the following just few point are reported:

the presentation is badly organised;

results and discussion are qualitatively described;

conclusions are too weak, and not strongly supported by the results;

english must be improved.

In my opinion the manuscript must be rejected.

Author Response

Dear Reviewer:
Thank you for reviewing the manuscript, we have made careful modifications to the manuscript in light of your valuable comments and suggestions. Please refer to the attachment for the modified content.

Thanks for your attention and patience. Hope you find this revised version satisfactory.

Reviewer 2 Report

This is an interesting article where the authors study a floating clamping action control technology.

The article is well written and illustrated. The main results of the article are of both scientific and practical interest. Experimental confirmation of one of the main results of the work, that the floating clamping action control strategy allows to reduce deformation during machining and increase the accuracy of the finished product, looks very convincing. Based on this, the reviewer recommends the article for publication, and also considers that the authors need to pay attention to the following aspects.

1. How efficient is such a process? What percentage of the material goes into chips. Can this process be compared to stamping\forging or 3D printing? Such questions should be considered at least briefly in the introduction.

2. What are the limits of applicability of this approach? Would it be desirable to be more specific about what wall thicknesses can be obtained? What materials meet these conditions? Etc.

3. Almost always, when it comes to optimizing real production processes, the technologist is faced with conflicting criteria. On the one hand, a decrease in energy of machining leads to an increase in quality, but on the other hand, this will probably also reduce productivity?

4. The conclusions are formulated at a qualitative level. Even in the abstract, the authors present some quantitative results. It is desirable to clarify the conclusions by adding specific data and results.

Author Response

(The authors gave the same response as above.)

Reviewer 3 Report

Dear authors,

In my opinion the subject is interesting, however you do not focus the interesting part the deformation after machining (the final deformation).

See comments in the annex.

Best regards,

Author Response

(The authors gave the same response as above.)

Reviewer 4 Report

The work proposes a manufacturing method specially designed for a part of the aeronautical sector, which as such requires the removal of 90% of the material. This may cause the material to deform to unacceptable values.

The work is interesting, it is well structured and it only requires to consider some minor issues:

- To include chemical composition and hardness of aluminum alloy

- To describe the fixation system principle: fixation force, hydaulic?

- It is necessary to describe the Renishaw measurement system

- Why were all the measurements made on the same machine and there was no initial and final measurement on a measuring marble?

- Shouldn't the boxes corresponding to the measurements be included in the process route (fig 10)

The authors are suggested to add an image in which the position of the 8 measurement points is indicated.

Author Response

(The authors gave the same response as above.)

Round 2

Reviewer 1 Report

The manuscript has nos not been improved.

Same considerations of V1: this is not a research article!

The manuscript must be rejected.

Reviewer 3 Report

Well done.